# Expansion of the Laser Beam Wavefront in Terms of Zernike Polynomials in the Problem of Turbulence Testing

**Alexey Rukosuev [1,\*], Alexander Nikitin [1], Vadim Belousov [1], Julia Sheldakova [1], Vladimir Toporovsky [1] and Alexis Kudryashov [1,2]**

1   Sadovsky Institute of Geosphere Dynamics of Russian Academy of Sciences, Leninskiy pr. 38/1, 119334 Moscow, Russia; nikitin@activeoptics.ru (A.N.); belousov@activeoptics.ru (V.B.); sheldakova@nightn.ru (J.S.); topor@activeoptics.ru (V.T.); kud@activeoptics.ru (A.K.)
2   Department of Physics, Moscow Polytechnic University, Bolshaya Semyonovskaya Str. 38, 107023 Moscow, Russia
\*   Correspondence: alru@nightn.ru

**Abstract:** The results of a study of the wavefront distortions of laser radiation caused by artificial turbulence obtained in laboratory conditions using a fan heater are presented. Decomposition of the wavefront in terms of Zernike polynomials is a standard procedure that traditionally is used to investigate the set of existing aberrations. In addition, the spectral analysis of the wavefront dynamics makes it possible to estimate the fraction of the energy distributed between different Zernike modes. It is shown that the fraction of energy related to the low-order polynomials is higher compared to the high-order polynomials. Also, one of the consequences of Taylor's hypothesis is confirmed—low-order aberrations are slower compared to the higher-order ones.

**Keywords:** adaptive optics; turbulence; wavefront sensor; Zernike polynomials; spectral analysis





## 1. Introduction

One of the most important applications of adaptive optics is the compensation of wavefront aberrations distorted by atmospheric turbulence. Recently, there has been interest in using the optical range for solving various problems: crypto-protected information transmission [1], organization of the optical communication channels in free space [2], recharging batteries of drones [3] low-orbit satellites [4], and destruction of space debris [5], etc. When a laser beam passes through the turbulent atmosphere of Earth, the quality of the wavefront (WF) corrupts, which leads to limitations in the operation of such systems. One of the methods of solving this problem is the use of an adaptive optical system (AOS), which makes it possible to correct for the WF in real-time. The frequency of correction is one of the main parameters of such a system, together with the number of corrected aberrations and their amplitudes. According to [6–8], under some "standard" weather conditions the frequency of the WF aberrations of the laser radiation distorted by the turbulent atmosphere rarely exceeds 100 Hz. Thus, the operating speed of a discrete adaptive optical system should be at least ten times higher—1 kHz [9,10].

The creation of an optical system operating in actual atmospheric conditions is quite costly and laborious. In addition, before entering the track, it is necessary to test the mathematical apparatus used in the research. Therefore, it was proposed to investigate the properties of the turbulence created in laboratory conditions.

## 2. Materials and Methods (Obtaining Experimental Data)

To analyze correctly the parameters of the wavefront distorted by turbulence, a sensor with the appropriate speed spatial resolution should be applied. The most suitable for this purpose is the Shack–Hartmann wavefront sensor (WFS) [11,12] since it can provide the required speed and high accuracy. To ensure the performance of the kilohertz range the high-speed camera JetCam-19 [13] was used [14], which, with an image resolution of

480 × 480, provided a frequency of 2000 frames per second. The WFS used an array of microlenses with the following parameters: the number of microlenses was 20 × 20, the focal length of the microlenses was 12 mm, and the pitch—240 μm.

The layout of the experimental setup is shown in Figure 1.

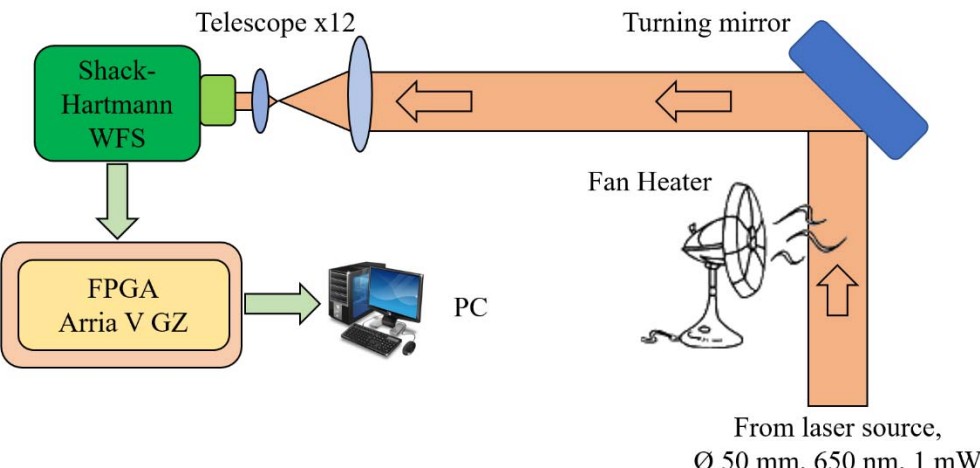

**Figure 1.** Experimental setup layout.

A diode laser beam with the wavelength of 650 nm and the power of 1 mW with the diameter of 50 mm passed through the moving heated airflow created by the fan heater. The airflow was directed perpendicular to the laser beam. To match the input aperture of the WFS and the beam aperture, a telescope with a magnification of ×12 was used.

Information from the WFS was processed in FPGA Arria V GZ, which calculated the coordinates of the focal spots in real-time (simultaneously with the reception of the image bytes). A turning mirror was used to reduce the size of the installation. If we want to correct for the laser beam wavefront, it should be replaced by a deformable mirror [15].

The rough data coming from the WFS is the displacement of the coordinates of the focal spots, formed by the images from the microlenses, relative to the coordinates of the reference wavefront (measured and loaded into the FPGA in advance) along X and Y axes. The displacement is proportional to the local slopes of the wavefront. The use of an array of the focal spots' displacements uniformly distributed over the beam aperture makes it possible to reconstruct the entire wavefront.

During the experiments, the set of frames, representing the displacements of the coordinates of the focal spots along *X* and *Y* axes, respectively, was recorded. With a sampling rate of 2 kHz and a recording time of 10 s, the total number of the stored frames was equal to 20,000. This ten-second recording made it possible to achieve a resolution along the frequency axis of 0.1 Hz [16]. The stored coordinates of the focal spots allowed for the reconstruction of the wavefront for each frame (hartmannogram).

To reconstruct the wavefront, the modal method described in [17] was applied. According to [18,19], Zernike polynomials are orthogonal in the unit circle. In our work, we used Wyant indices of Zernike polynomials [20] (Z1 and Z2—tilts, Z3—defocus, Z4 and Z5—astigmatism, Z6 and Z7—coma, Z8—spherical, etc.). A total of 24 polynomials was used for wavefront decomposition.

The wavefront $\Phi(x, y)$ of laser radiation falling on the WFS can be presented as a linear combination of polynomials:

$$\Phi(x, y) = \sum_{i}^{m} c_i Z_i(x, y) \tag{1}$$

where $(x, y)$—focal spot coordinates,

$m$—the number of polynomials involved in the expansion, strictly speaking, this number should be equal to infinity,

$c_i$—coefficients of Zernike polynomials,
$Zi(x,y)$—the value of Zernike polynomial at the given point $(x, y)$.
It is assumed that all physical coordinates are normalized to the unit circle.
The focal spots' shifts measured by the sensor can also be written as:

$$\Delta S_x = f\frac{\partial \Phi(x,y)}{\partial x} = \sum_i^m c_i X_i(x,y) \tag{2}$$

$$\Delta S_y = f\frac{\partial \Phi(x,y)}{\partial y} = \sum_i^m c_i Y_i(x,y) \tag{3}$$

where

$$X_i = f\frac{\partial Z_i(x,y)}{\partial x}, \ Y_i = f\frac{\partial Z_i(x,y)}{\partial y} \tag{4}$$

$f$—the distance between the lens array and the camera sensor of the WFS.
The expansion coefficients $c_i$ are calculated from the minimization of the functional:

$$\chi^2 = \sum_j^n \left\{ \left[\Delta S_{xj} - \sum_i^m c_i X_i(x_j, y_j)\right]^2 + \left[\Delta S_{yj} - \sum_i^m c_i Y_i(x_j, y_j)\right]^2 \right\} \tag{5}$$

where $n$—the focal spot number
by solving the system of equations:

$$\frac{\partial \chi^2}{\partial c_i} = 0 \tag{6}$$

in matrix presentation:

$$Mc = S \tag{7}$$

where

$$M = X^T X + Y^T Y \tag{8}$$

$$X = \begin{bmatrix} X_1(x_1, y_1) & \dots & X_m(x_1, y_1) \\ \dots & \dots & \dots \\ X_1(x_n, y_n) & \dots & X_m(x_n, y_n) \end{bmatrix}, \ Y = \begin{bmatrix} Y_1(x_1, y_1) & \dots & Y_m(x_1, y_1) \\ \dots & \dots & \dots \\ Y_1(x_n, y_n) & \dots & Y_m(x_n, y_n) \end{bmatrix} \tag{9}$$

$$c = \begin{bmatrix} c_1 \\ \dots \\ c_m \end{bmatrix} \tag{10}$$

$$S = X^T s_x + Y^T s_y \tag{11}$$

$$s_x = \begin{bmatrix} \Delta S_{x_1} \\ \dots \\ \Delta S_{x_n} \end{bmatrix}, \ s_y = \begin{bmatrix} \Delta S_{y_1} \\ \dots \\ \Delta S_{y_n} \end{bmatrix} \tag{12}$$

$X^T$ and $Y^T$ are transposed matrices of $X$ and $Y$.
Thus, the expansion coefficients $c_i$, which give an idea of the magnitude of various types of wavefront aberrations, are calculated from

$$c = M^{-1}S \tag{13}$$

using the singular value decomposition of the matrix $M$.

## 3. Results

The representation of the wavefront in terms of Zernike polynomials obtained using such a procedure was saved as a new sequence of frames. Thus, we have the sequence of wavefronts presented in terms of Zernike polynomials separated by intervals of 500 μs.

It is possible to obtain the spectrum for each Zernike polynomial by using the discrete Fourier transform [21]. Figure 2 represents the time dependence of the defocus aberration (Zernike polynomial #3). For a more detailed demonstration of the changes in magnitude, a sample fragment with a length of approximately 7 ms is shown.

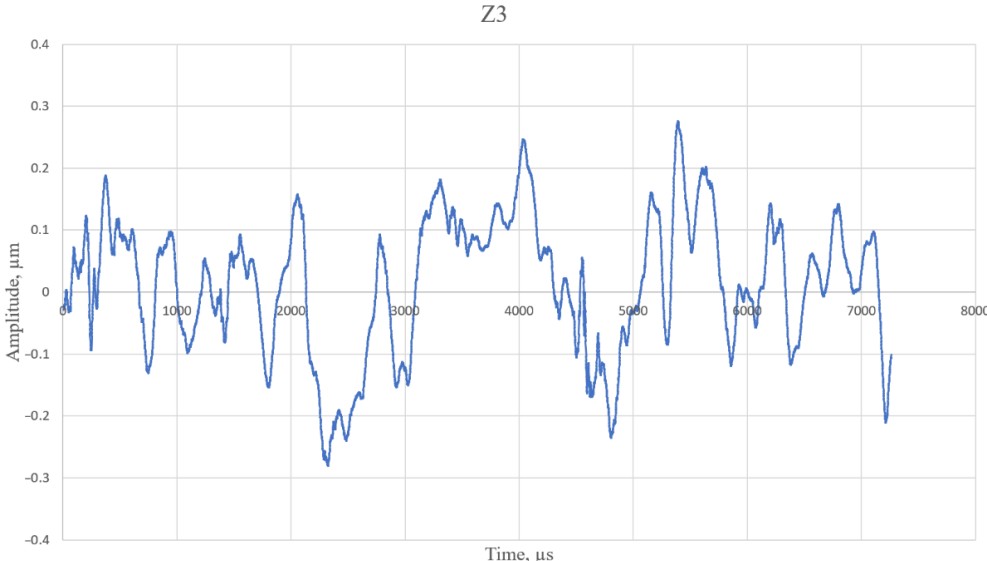

**Figure 2.** Time sampling for Zernike coefficient #3 (defocus).

Figure 3 shows the defocus power spectral density. For this graph, a portion of the sample size of 4096 elements was used to increase the speed of calculations. That gave a resolution along the frequency axis of approximately 0.5 Hz.

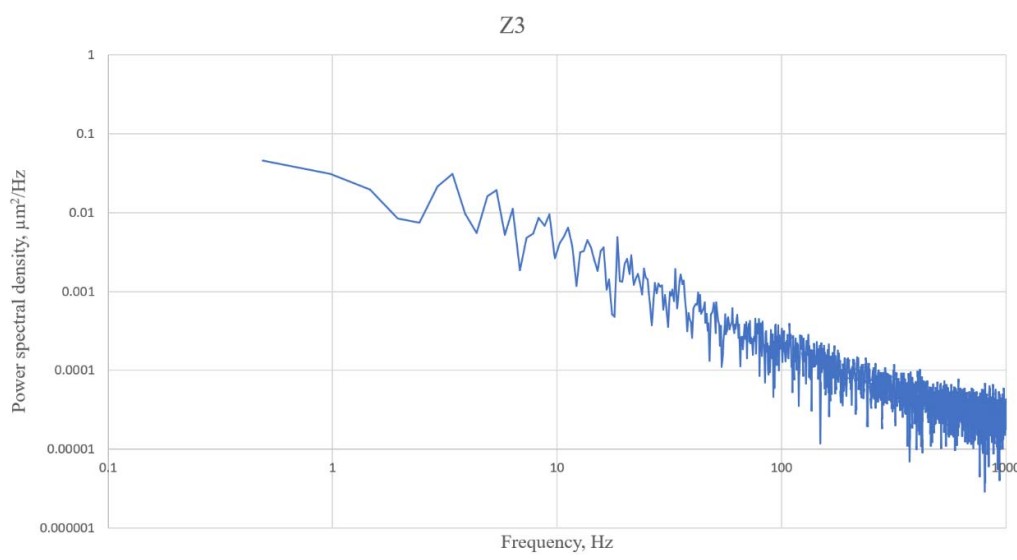

**Figure 3.** Power spectral density for Zernike coefficient #3 (defocus).

After taking the integral of the spectral power density, we obtained some analogs of energy for each aberration. The concept of the energy cannot be strictly defined in relation to the phase, however, the formulas we used to calculate this value are similar to the formulas for energy calculation. Nevertheless, since there is a direct relationship between turbulence and the distortion of the wavefront (the stronger the turbulence, the greater the distortion of the wavefront), this value, according to our opinion, can characterize the real energy of the turbulent airflow. For this reason, in the text of the article, we shall use this term in quotation marks—"energy". Figure 4 presents the results of integrating the power spectral density data obtained for the first eight Zernike polynomials.

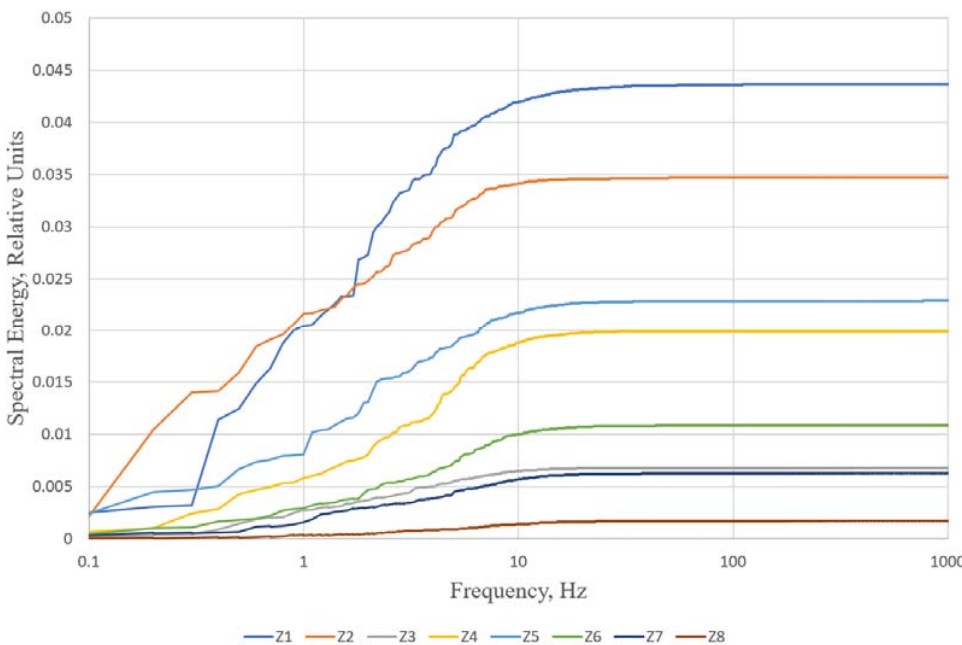

**Figure 4.** Spectral energy for first eight Zernike polynomials.

The graphs in Figure 4, upon reaching some frequency, come to saturation, which indicates the absence of any significant contribution to the amplitude of the high-frequency components. The frequency at which saturation occurs can be taken as the bandwidth of different polynomials (Figure 5).

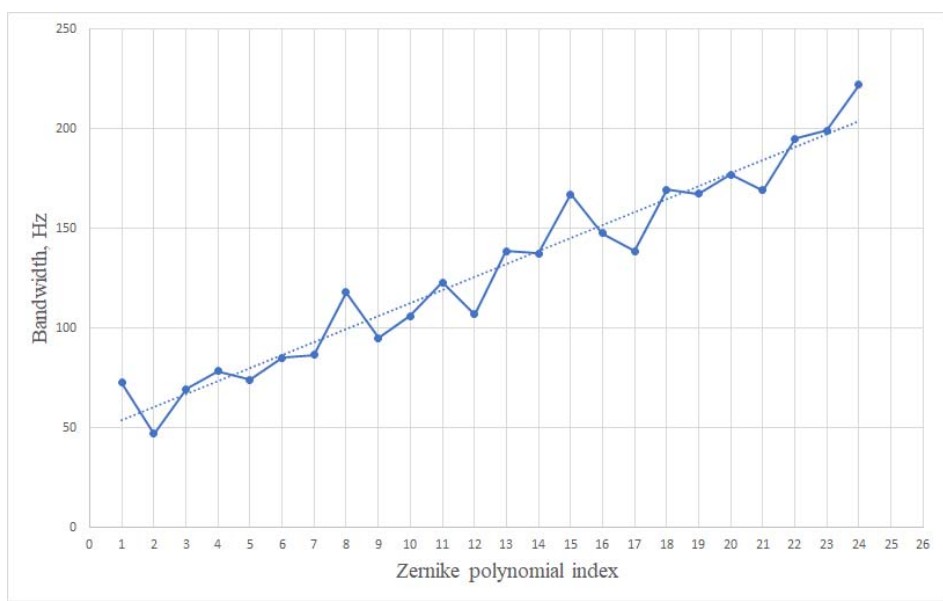

**Figure 5.** Bandwidth of the first 24 Zernike polynomials.

Thus, one of the consequences of Taylor's hypothesis [22], according to which low-order aberrations have a lower frequency compared to high-order aberrations, is confirmed. This can be explained by the fact that in our experiments, the fan heater was installed at a short distance from the laser beam. The warm air jet, among other things, was reflected from the cold optical table and from the screen that blocked the access of the airflow to other elements of the optical circuit. When mixing with the cold air of the laboratory, the heat flow formed turbulent vortices. In particular, the Fried parameter was determined, which turned out to be equal to 10 mm. Apparently, for these reasons, in our experiments, the airflow was more turbulent than laminar, and the Taylor hypothesis was generally followed.

On the other hand, it is possible to build a graph that reflects the saturation amplitudes for each coefficient $c_i$ (Figure 6).

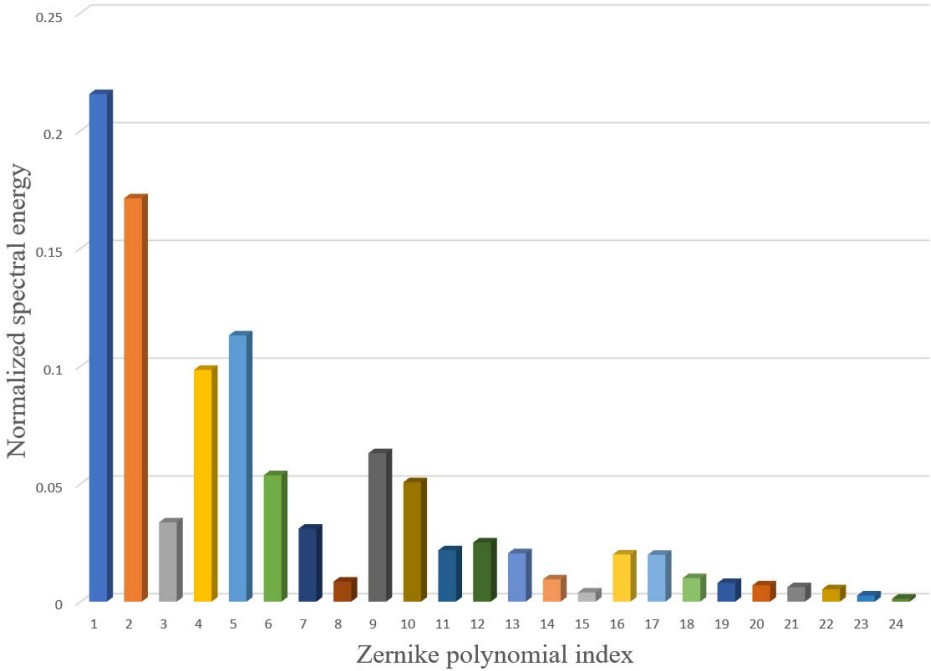

**Figure 6.** Distribution of turbulence energy over Zernike polynomials.

From the examination of this figure, it can be seen that defocus (Z3) and spherical aberrations (Z8, Z15, and Z24) have a smaller amplitude compared to the rest of the aberrations (coma, astigmatism). In general, from the experimental point, it is understandable as the flue of the heated air is perpendicular to the direction of the beam distribution and thus the asymmetry should be observed. This information should be useful in optimizing the spatial arrangement of the wavefront corrector control elements.

If you take the integral of the values shown in Figure 6, then you can obtain a graph displaying the integral energy concentrated in the first N Zernike polynomials (Figure 7).

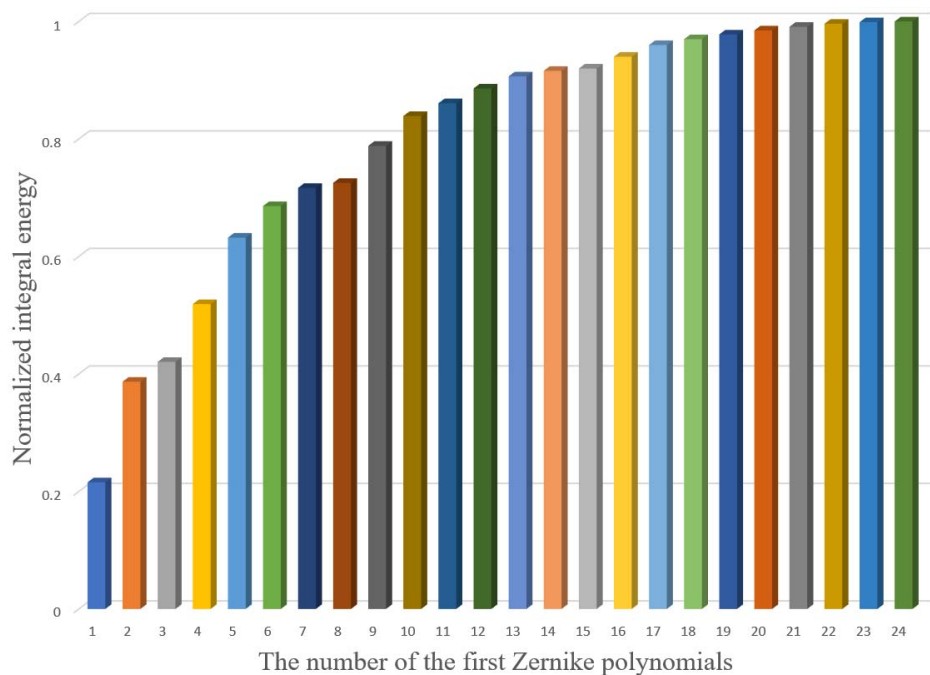

**Figure 7.** Normalized turbulence integral energy over first N Zernike polynomials.

It can be seen from this graph that, for example, tilts contain approximately 40% of all phase fluctuation "energy". The first eight polynomials (tilts, defocus, astigmatism, coma, and spherical aberration) contain approximately 70%, while the first 15 polynomials contain approximately 90% of all energy.

## 4. Discussion

Fourier analysis, carried out in conjunction with the expansion of the wavefront in terms of Zernike polynomials, makes it possible to determine the statistical characteristics for each aberration corresponding to a particular polynomial. This could be useful when designing adaptive optical systems, particularly when choosing the system speed and spatial resolution of the wavefront corrector actuators. For example, if the system can correct for the aberrations of the first 15 Zernike polynomials, then up to 90% of the WF distortions introduced by turbulence will be compensated. In that case, probably, it would be useful to apply the low-cost bimorph deformable mirror that can successfully reproduce the first 25 Zernike modes [23,24]. It can also be noted that the created mathematical apparatus will make it possible to analyze the parameters of real atmospheric turbulence.

**Author Contributions:** Conceptualization, A.K., A.R.; methodology, A.K., A.R.; software, V.B., A.R.; validation, A.K., A.N., J.S.; formal analysis, A.N., A.R., V.T.; investigation, A.R., A.N., V.B.; resources, A.K.; data curation, A.R., V.T.; writing—original draft preparation, A.R.; writing—review and editing, A.N., J.S., V.T.; visualization, A.N., A.R.; supervision, A.K.; project administration, A.K. All authors have read and agreed to the published version of the manuscript.

**Funding:** This research was funded by the RUSSIAN SCIENCE FOUNDATION, grant number 19-19-00706.

**Institutional Review Board Statement:** Not applicable.

**Informed Consent Statement:** Not applicable.

**Data Availability Statement:** The data presented in this study are available on request from the corresponding author. The work is supported by the Russian Science Foundation under grant #19-19-00706.

**Conflicts of Interest:** The authors declare no conflict of interest. The funders had no role in the design of the study; in the collection, analyses, or interpretation of data; in the writing of the manuscript, or in the decision to publish the results.

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
