# Peer review of "Expansion of the Laser Beam Wavefront in Terms of Zernike Polynomials in the Problem of Turbulence Testing"

_applsci, doi:10.3390/app112412112_

Round 1

Reviewer 1 Report

In order to give full play to the performance of adaptive optic system, it is necessary to understand the characteristic of atmospheric turbulence in application scenarios. In this article, the Zernike polynomials expansion of laser beam wavefront in turbulence testing is developed, which has important application value for adaptive optics system design.

Firstly, the wavefront of the laser beam caused by artificial turbulence generated by a fan heater in the laboratory is decomposed into Zernike polynomials. Secondly, the statistical characteristics for each aberration corresponding to a particular polynomial are analyzed and the energy distribution of different Zernike modes is estimated.

In a word, this article is innovative.

Author Response

We would like to thank our reviewer for the time and efforts on this peer-review process.

Reviewer 2 Report

This paper present the results of an experiment on turbulence testing. I think the paper can be accepted with minor changes, but I suggest the authors to study more in depth the data acquired and make some comparisons with experimental and theoretical results of the existing literature.

I think that sections 3 and 4 contains very few citations. I suggested a few below in the detailed comments, but I suggest to the authors to search also for others.

My detailed comments.

  • line 32: I think a citation for the sentence "turbulent atmosphere rarely exceeds 100 Hz" is required. I suggest Conan J.-M. et al. JOSA A 1995
  • line 86, "ci" i should be a subscript
  • line 88, I think this line should be moved just after or before eq (5)
  • line 104 and figure 2. Temporal length of the variable reported here does not correspond to "With a sampling rate of 2 kHz and a recording time of 10 s, the total number of the stored frames was equal to 20,000" (line 65-67). I think some information should be added to explain this discrepancy. 
  • Same point for line 107 and figure 3. I expected a value at 0.1Hz (see "This ten-second recording made it possible to achieve a resolution along the frequency axis of 0.1 Hz" line 67-68)
  • Line 108-109 "we get some analog of energy for each aberration". This sentence is not clear to me, but I understand what you meant in the following sentence.
  • Line 109-110. "The concept of the energy cannot be strictly defined in relation to the phase". I think you can develop this point in more details. There is a relationship between refractive index fluctuations (optical turbulence) and the variance of the phase. I suggest you these publications: Noll JOSA A 1975,  Roddier Progress in Optics 1981 and Roddier "Adaptive optics in Astronomy" 1999.
  • Line 121 and 122: "The frequency at which saturation occurs can be taken as the bandwidth of different polynomials (Figure 5)." I think a more significative and sound bandwidth is the frequency for a magnitude of -3dB. Then, I think a comparison with Conan J.-M. et al. JOSA A 1995 (or an analogous publication) will be interesting.
  • Line 128 "ci": subscript.
  • Line 151, 152: "then up to 90% of the WF distortions introduced by turbulence will be compensated. " I think you can compare this result with same results in literature. Just a couple of examples are Noll JOSA A 1975 and Roddier "Adaptive optics in Astronomy" 1999

Author Response

We would like to thank our reviewer for the time and efforts on this peer-review process. The comments are really important. Please see the attachment.

Reviewer 3 Report

The manuscript presents the results of laboratory experiment to measure wavefront aberrations using a classical Shack-Hartmann wavefront-sensor.  The experiment uses a fan and an heating element to create turbulence in the optical path of a laser beam and these aberrations are then measured and evaluated in terms of a Zernike polynomial expansion. All this is well understood and well documented in the literature. The authors confirm the Tylor's frozen atmosphere hypothesis. This is a bit surprising result that I would encourage the author to explain better. The use of a fan and heating element would, in principle, generate a more laminar flow situation. In such physical conditions it's then surprising that the power spectrum of the aberrations  follows such hypothesis. 

Author Response

We would like to thank our reviewer for the time and efforts on this peer-review process. The comments are really important. Here is our response to comments:

In our experiments, the fan heater was installed at a short distance from the laser beam. The warm air jet, among other things, was reflected from the cold optical table and from the screen that blocked the access of the airflow to other elements of the optical circuit. Mixing with the cold air of the laboratory, the heat flow formed turbulent vortices. In particular, the Fried parameter was determined, which turned out to be equal to 10 mm. Apparently, for these reasons, in our experiments, the airflow was more turbulent than laminar, and Taylor’s hypothesis was generally followed.